# Identification of Potential Therapeutic Targets on the Level of DNA/mRNAs, Proteins and Metabolites: A Systematic Mapping Review of Scientific Texts' Fragments from Open Targets

**Pavel V. Pogodin *** , **Olga I. Kiseleva** and **Ekaterina V. Ilgisonis**

Institute of Biomedical Chemistry, Pogodinskaya Street, 10, 119121 Moscow, Russia
* Correspondence: pogodinpv@gmail.com

**Abstract:** Database records contain useful information, which is readily available, but, unfortunately, limited compared to the source (publications). Our study reviewed the text fragments supporting the association between the biological macromolecules and diseases from Open Targets to map them on the biological level of study (DNA/RNA, proteins, metabolites). We screened records using a dictionary containing terms related to the selected levels of study, reviewed 600 hits manually and used machine learning to classify 31,260 text fragments. Our results indicate that association studies between diseases and macromolecules conducted on the level of DNA and RNA prevail, followed by the studies on the level of proteins and metabolites. We conclude that there is a clear need to translate the knowledge from the DNA/RNA level to the evidence on the level of proteins and metabolites. Since genes and their transcripts rarely act in the cell by themselves, more direct evidence may be of greater value for basic and applied research.

**Keywords:** therapeutic targets discovery; textual evidence of a target-disease association; machine learning; text analysis; DNA and RNA; proteins; metabolites

## 1. Introduction

Therapeutic targets are biomacromolecules whose behavior could be modulated by the active ingredient of medication to correct a certain pathological state [1]. Therefore, the discovery and validation of novel therapeutic targets are essential, since novel target means, new therapeutic opportunities are needed to treat diseases more efficiently and address still unmet medical needs. Thus, it is unsurprising that numerous studies aim to identify the potential therapeutic targets among biomacromolecules, primarily proteins [2], in the human body. Also, the results of such studies are available en masse through dedicated internet resources such as Open Targets [3] and several others [4,5].

One of the essential requirements for a biomacromolecule to serve as the therapeutic target is its involvement in disease-related processes. Causal relationships and associations between biomacromolecules and diseases are studied extensively using different experimental methods on the level of DNA/RNA, proteins and metabolites and are covered by the scientific literature. In turn, scientific literature may be processed and summarized using automated or semi-automated approaches to provide the overall association score between the particular disease and the biological entity (biomacromolecule). Resources like Open Targets provide such information for the researchers, who can use it to prioritize potential targets and plan their biomedical studies, benefiting from the existing collective knowledge on the subject without an independent extensive literature search.

Nevertheless, information loss inevitably occurs during the process of summarization. In the case of literature-based judgment on the association between a biological entity and disease in Open Targets, it occurs too: data on the biological level of study is absent.

I.e., it is not explicitly stated what exactly was studied in the context of disease: gene and its variants (DNA), transcript (mRNA), a final gene product (protein), or related organic compound (metabolite).

In our study, we applied a well-established systematic review methodology to map [6,7] the scientific texts' fragments supporting the association between biological entities and diseases from Open Targets onto the biological levels of studies (DNA/mRNA level, protein level and metabolite level). We did it to assess the prevalence of different biological levels of studies in the search for potential therapeutic targets. We presumed that our results would highlight the gaps in current knowledge on the subject and provide the means to narrow them. Instead, our results indicate the prevalence of genomic and transcriptomic evidence for the role of distinct biological macromolecules in pathology. Since genes and their transcripts rarely act in the cell by themselves and more direct evidence may be of greater value for the basic and applied studies of potential therapeutic targets, there is a clear need to translate the DNA/RNAs evidence into the ones on the level of proteins and metabolites [2,8].

## 2. Methods

This systematic review was conducted under the Preferred Reporting Items for Systematic Reviews and Meta-Analyses (PRISMA) guidelines [9], plan of the study was pre-registered in Open Science Framework (OSF) [10] (https://osf.io/9w5x3/?view_only=f4fee9d348a2406fa3d2e0e9c75f83ff (accessed on 3 December 2022)), registration number is h82a4. It is important to note that not all scientific articles had been manually reviewed and mapped, but only their fragments contained in the materials of the Open Targets platform [3]. We undertook such an approach given the large scale of the theme under review. Moreover, in addition to the traditional manual review, we used machine learning text classification to cover all the text fragments objectively included in this study. The overview of the research is given in Figure 1. The following text describes the materials and methods used in this study in more detail.

**Text fragments from Open Targets.** Open Targets platform (https://platform.opentargets.org/ (accessed on 3 December 2022)) is a comprehensive open-source research tool that supports systematic identification and prioritization of potential therapeutic targets for drug discovery and development [3]. The main advantage of the Open Targets Platform is its openness. The platform provides transparency in the drug discovery process by making the data and analysis available to the whole community, including academic researchers, pharmaceutical companies, and technology experts. This openness allows for greater collaboration from the scientific community, leading to more reliable results. Another significant advantage of the Open Targets platform is its multidisciplinary. The platform integrates various types of data to provide a comprehensive understanding of the machinery of a disease.

Furthermore, machine learning algorithms of Open Targets allow one to prioritize potential drug targets based on their druggability. It focuses the resources of the scientists on the most promising targets. The main disadvantages of the Open Targets platform are the non-consistency of the quality and completeness of the data accumulated and biases of the machine learning algorithms towards certain types of targets or diseases. Nevertheless, the Open Targets platform is a valuable solution to drug development, providing a transparent and systematic approach to identifying and validating drug targets. In the context of this study, it is important that Open Targets connects biological molecules (targets) and diseases and contains supporting evidence, including those extracted from the scientific literature. For each such piece of evidence, target ID, target name and a specific article fragment connecting this target and the disease are available. Our study used Open Targets v. 21.04 (https://ftp.ebi.ac.uk/pub/databases/opentargets/platform/21.04/output/etl/json/evidence/sourceId=europepmc/, accessed on 3 December 2022), since, in contrast with the newer versions, it contains targets' names as they are given in the corresponding text fragments, which was helpful in our analysis.

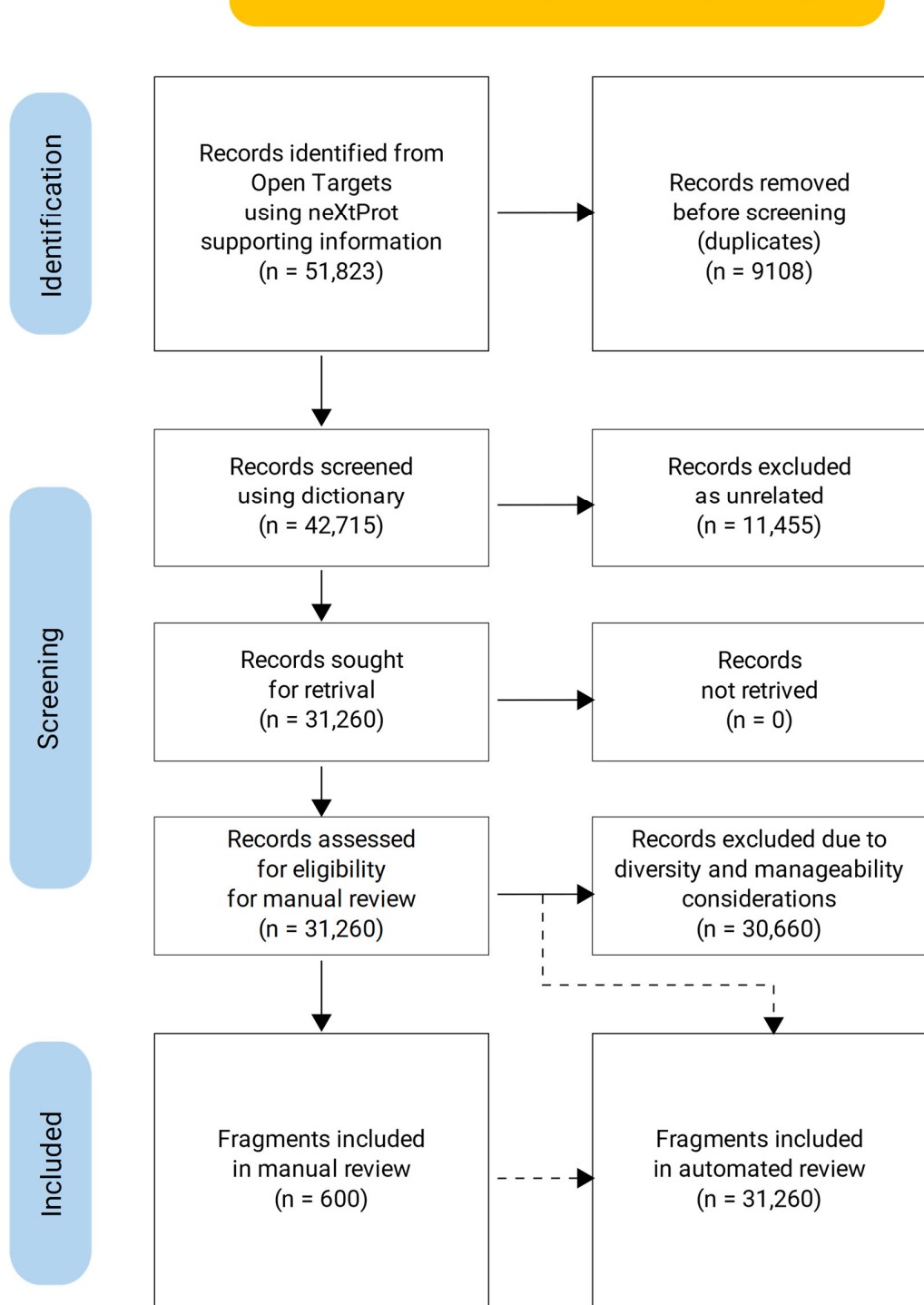

**Figure 1.** PRISMA diagram describing the data flow in the systematic review process. Text fragments from Open Targets underwent several stages of processing. First, records identified from Open Targets were filtered out from duplicates (ca. 17%) and screened using a dictionary. After that, less than 2% of the unique entries (manageable subset) were reviewed manually using the in-house tool TCSTF. Finally, all unique entries were subjected to automated review using a naïve Bayes classifier built using the manual review results (dotted arrows).

**Supporting information on biological entities from neXtProt to identify candidate text fragments.** Data on proteins, their biologically relevant properties, and corresponding evidence were obtained from neXtProt (https://www.nextprot.org/, accessed on 4 December 2022). The neXtProt knowledge base is an integrative resource providing rich data on human proteins [11]. In the context of this study, it is important that neXtProt contains annotations of proteins, which correspond to the various functions and properties proteins may have. Each annotation has its quality level (Bronze, Silver, and Gold) and list of evidence. This evidence also has a quality level (Bronze, Silver, and Gold) and the source, including but not limited to the particular publication. We used this information as the filtering criteria to include in our further analysis only those records from Open Targets, which correspond to the articles listed as high-quality evidence for high-quality annotations. The current version of neXtProt was downloaded on 12 December 2022 in XML format.

Criteria for the selection of the records on biological entities and articles to identify the candidate text fragments in Open Targets:

1. The biological entity has evidence of existence at the protein level.
2. Annotation belongs to the following hierarchy groups: (a) general-annotation; generic-function, (b) general-annotation, (c) general-annotation; generic-interaction, (d) general-annotation; medical.
3. The quality of annotation is designated as "GOLD".
4. The annotation does not belong to the following categories (a) caution, (b) domain-info, (c) sequence-caution, (d) variant-info.
5. Annotation has no negative evidence.
6. Quality of evidence is designated as "GOLD".
7. The type of evidence is "publication".
8. Terms describing the evidence are (a) physical interaction evidence used in manual assertion, (b) experimental evidence used in manual assertion, (c) direct assay evidence used in manual assertion, and (d) experimental evidence.
9. PubMed ID exists for the publication.
10. neXtProt ID for the biological entity has a one-to-one relationship to the Ensemble gene ID, which is used in Open Targets.

**Identification of the candidate text fragments.** We selected only those records from Open Targets data, which were backed by publications supporting protein annotations in the processed neXtProt data. We used PubMed IDs of the articles and Ensemble gene IDs to pool the data from Open Targets and neXtProt.

**A dictionary containing search terms and screening.** Freely available recommendations on the terminology of methods of bioanalytical chemistry from IUPAC [12] were used to create a dictionary. Terms were extracted from the aforementioned book [12] along with their categories: Enzymatic methods, Immunoanalytical methods, Genomics and Nucleic Acid Analysis, Proteomics, Metabolomics, Glycomics, and Lipidomics. In our study, we grouped terms into three broader categories:

1. Terms describing experimental studies conducted on the nucleic acids (DNA or RNA) level.
2. Terms describing experimental studies conducted on the protein level.
3. Terms describing experimental studies conducted on the metabolite level.

Screening of text fragments was conducted based on the presence of the term from a specific category in the text fragment. That is when the fragment contained the term describing experimental studies on the protein level. The fragment was counted as a member of the corresponding category. Our screening strategy allowed the terms to be a part of the longer words. However, we make an exception for the several short terms: they were counted only as single words since they were useless as a part of the whole, longer unspecific words (examples: $\backslash b$ria$\backslash b$, $\backslash b$gag$\backslash b$, where "$\backslash b$" stands for the word

boundaries.). All search terms, along with the corresponding categories, can be found in Supplementary Table S1.

**Manual review of the text fragments.** Two hundred text fragments from each previously dictionary-defined category (DNA/RNA level, protein level, metabolite level) were selected to maximize their diversity concerning detected terms and overall text similarity. The text fragments were selected by minimizing the total sums of cosine similarities for whole text fragments and terms found in them.

Detected terms related to the experimental method and names of the biological entities were highlighted in those text fragments. This could be illustrated by the following example, where the name of the biological entity is given in square brackets, and terms related to the level of study are presented in curvy brackets:

"Using IP with an anti-acetyl lysine {antibody}, we identified [Aurora B] as an acetylated {protein} in PC3 prostate cancer cells. [Aurora B] is regulated by acetylation/deacetylation during mitosis in prostate cancer cells".

Text fragments were loaded into the in-house developed Tool for Categorization of Short Text Fragments, TCSTF [13]. TCSTF was developed to simplify the fascinating but laborious text analysis and categorization (labeling) procedure, which will help the researchers model, predict, and explain biomedical phenomena better. TCSTF represents a simple web form containing formatted text fragments and three questions to answer:

1. Does the text fragment allow us to judge the category of the experimental method?
2. Are highlighted terms sufficient to judge the category of experimental method?
3. Is the specific level of the study mentioned in the text fragment?

Answering these three questions allows us to judge the level of study (question 3) and get the characteristics of the text fragments and terms found in the text fragment (questions 1 and 2, respectively).

Using this tool, the belonging to the previously defined category was validated or invalidated by the consensus decision of the authors of this paper (see Figure 2 for the example). TCSTF with the embedded text fragments may be found in the Supplementary Materials (see Supplementary Files S1–S3).

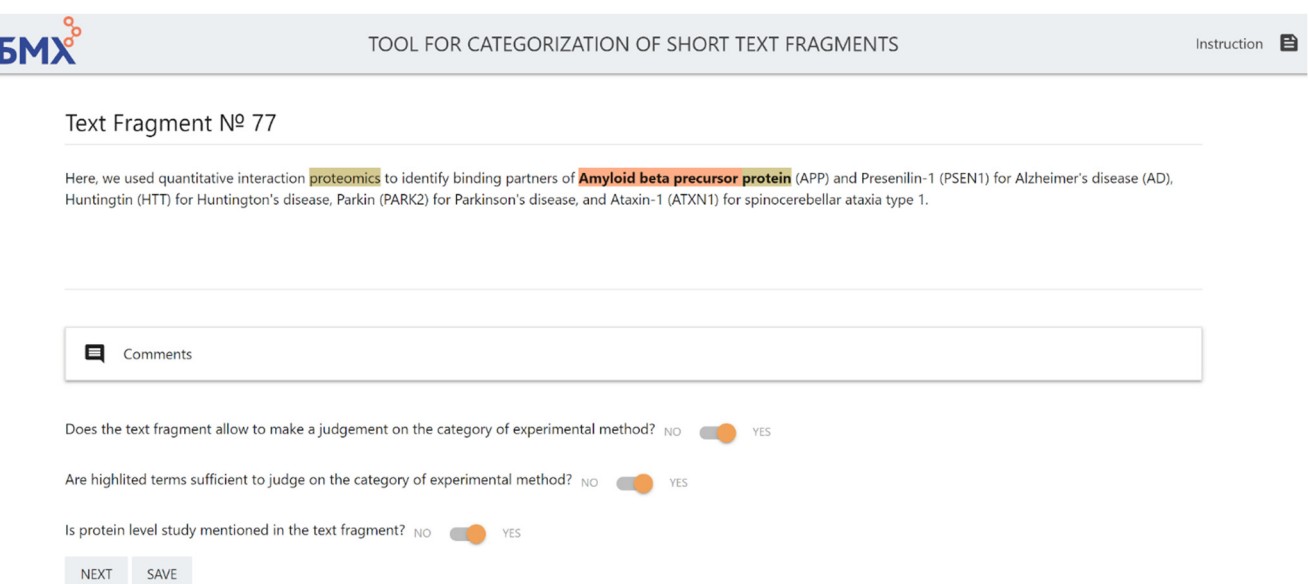

**Figure 2.** Manual review of text fragments using TCSTF. Names of the biological entities and terms from the dictionary are highlighted in color.

**Automated review of the text fragments.** Naive Bayes classifiers were built using the results of the manual classification as the training sets: text fragments were tokenized by distinct words; fragments with validated belonging to the category were used as positive

examples, others—as negative ones. Quality assessment of the classifiers was conducted during the 5-fold Cross-validation. Balanced Accuracy varied from 0.70 (category metabolites) to 0.73 (category proteins). Classifiers were used to predict the category for all the fragments which previously had passed dictionary-based screening.

**Technical realization.** Open Targets data were downloaded and processed using R-scripts [14] written using the functionality of such packages as RCurl [15], jsonlite [16] and tidyverse [17]. Data from neXtProt were downloaded manually and processed using PHP, its extension XMLReader (https://www.php.net/manual/en/intro.xmlreader.php, accessed on 9 April 2023) tidyverse [18]. R package kofnGA [19] was used to select a diverse subset of text fragments for the manual analysis. R packages from the quanteda family were used to process text fragments [20]. R package caret was used to build classifiers [21]. R packages ggplot2 [22] and diagrammeR [23] were used to prepare plots, and a flow diagram describing the process of the review was created using Inkscape software (https://inkscape.org/, accessed on 9 April 2023).

## 3. Results and Discussion

### 3.1. Dictionary-Based Screening

Dictionary-based screening allowed us to elucidate 31,260 text fragments presumably describing the experimental studies on DNA and RNA, protein, and metabolite levels (see Figure 3).

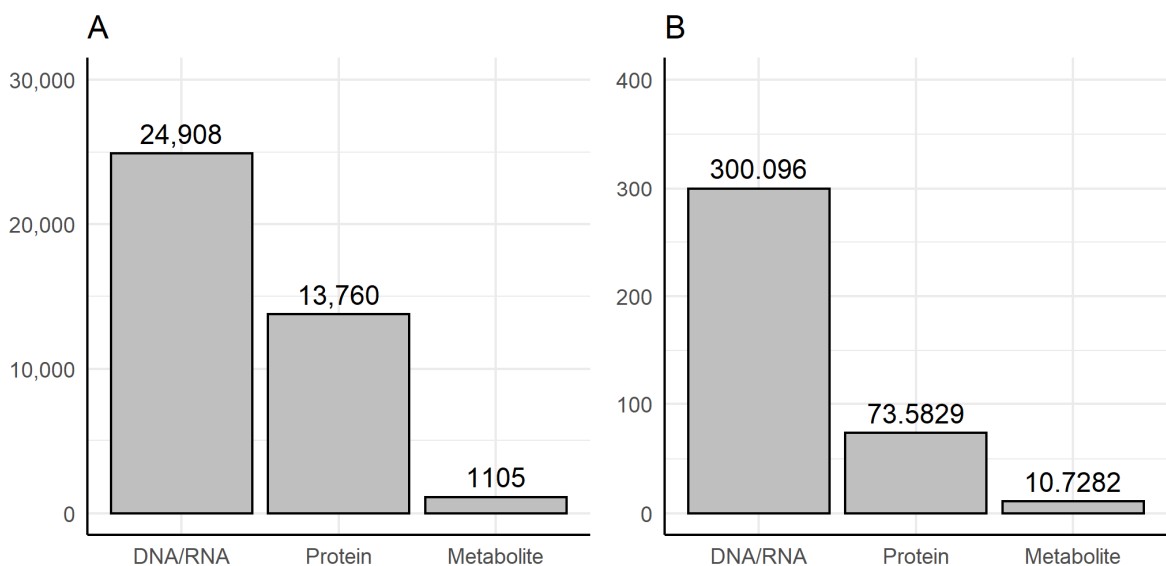

**Figure 3.** Results of the dictionary-based screening. (**A**)—raw numbers of the text fragments in categories. (**B**)—numbers adjusted by the dictionary's size (raw number/number of terms in the corresponding category).

These results align with the current view of biomedical studies: analyzing nucleic acids is probably the most accessible and high-throughput option since polymerase chain reaction (PCR) allows one to study small quantities of biological material fast with affordable equipment [24]. Working with proteins is usually more demanding since there is no convenient way to multiply them in vitro [25]. Moreover, many proteins in varying concentrations form a complex mixture in the cell [26], which is hard to analyze quantitatively and qualitatively without expensive and intricate technological setups [27]. Analysis of the metabolite level also may be technically convoluted. Besides, it demands preexisting knowledge on the connection between metabolites and particular genes/proteins, and that limits the number of genes/proteins for which the association with the disease may be drawn through the metabolite.

Given that our dictionary has different numbers of terms in each category, we adjusted the results given in Figure 3A by the number of terms in corresponding dictionary cate-

gories. As seen in Figure 3B, the results did not change fundamentally: the largest number of text fragments is still associated with the DNA/RNA level, and the smallest number is with the metabolite level. The screening results obtained using a simple dictionary seem legit from the commonsense perspective.

### 3.2. Manual Review of the Selected Text Fragments

Using the results of the dictionary-based screening, we selected a small but diverse subset of the text fragments (two hundred in each previously defined category) and reviewed them manually. The results are depicted in Figure 4.

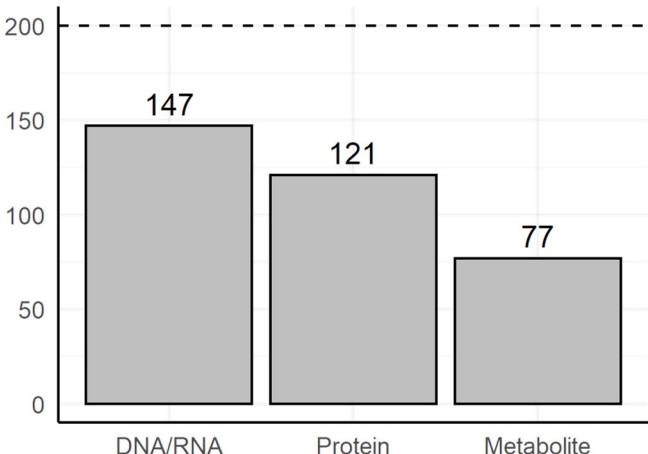

**Figure 4.** Results of the manual review of 200 text fragments in each category.

As can be seen from Figure 4, far from everything was verified during the manual review: from 26 to 61 percent of the text fragments were false positives of the dictionary-based screening. These results indicate that dictionary-based screening of the text fragments is too simple and cannot adequately handle the heterogeneous text fragments and provide us with reliable final mapping. On the positive side, at this step of the study, we manually mapped a small fraction of the text fragments, which can be used as the training sets needed to apply more sophisticated and, hopefully, more reliable approaches for the automated review of the text fragments. The results of the manual mapping review may be found in the Supplementary Materials (see Table S2), examples of the text fragments, which have been reviewed positively, are given in Table 1 (5 randomly selected examples from each dictionary-defined category).

**Table 1.** Examples of text fragments belonging to the dictionary-defined category were validated during the manual review.

| Number of the Text Fragments in TCSTF | Biological Entity | Text Fragment | Category |
|---|---|---|---|
| 72 | FTL | In our patients with undetectable serum ferritin levels, idiopathic generalized seizures, and atypical RLS, we screened all exons encoding the FTL gene for neuroferritinopathy-associated mutations. As a result, we detected a G > T nucleotide substitution (G310T) in exon 3 (Figure 1A). | DNA/RNA level |
| 1 | DOCK8 | Combined immunodeficiency associated with DOCK8 mutations. | DNA/RNA level |
| 3 | IL23R | A genome-wide association study identifies IL23R as an inflammatory bowel disease gene. | DNA/RNA level |
| 153 | GLI2 | We describe three novel heterozygous frameshift or nonsense GLI2 mutations, predicting truncated proteins lacking the activator domain associated with IGHD or combined pituitary hormone deficiency and ectopic posterior pituitary lobe without HPE. In addition, the GLI2 coding region of patients with isolated GH deficiency (IGHD) or combined pituitary hormone deficiency was amplified by PCR using intronic primers and sequenced. | DNA/RNA level |

**Table 1.** *Cont.*

| Number of the Text Fragments in TCSTF | Biological Entity | Text Fragment | Category |
|---|---|---|---|
| 90 | IDH2 | Subsequent studies revealed that IDH mutations were extremely rare in primary (de novo) glioblastomas but were common in recurrent glioblastomas developing secondary to low-grade tumors, which frequently have IDH mutations [5]. These findings suggested that IDH mutations are an early event in gliomagenesis and persist during the progression to recurrent glioblastomas. In addition, somatic heterozygous IDH1 or IDH2 mutations have frequently been detected in glioma/glioblastomas by genome-wide mutation searches [3,4]. | DNA/RNA level |
| 184 | SRPK1 | Lysates were separated by SDS-PAGE, transferred to nitrocellulose, and subjected to Western blot analysis using anti-SRPK1 (A), anti-SRPK2 (B), anti-topoisomerase I: arthritis foundation/CDC reference sera (C), and anti-cdc2 (CDK1) (mouse monoclonal) (D). | Protein level |
| 152 | Spexin | Spexin is a Novel Human Peptide that Reduces Adipocyte Uptake of Long Chain Fatty Acids and Causes Weight Loss in Rodents with Diet-induced Obesity Spexin is a novel human peptide that reduces adipocyte uptake of long-chain fatty acids and causes weight loss in rodents with diet-induced obesity. Spexin is a novel hormone involved in weight regulation, potentially for obesity therapy. A commercial immunoassay allowed us to examine possible relationships between circulating levels of Spexin and those of known obesity-related adipokines in human sera. | Protein level |
| 123 | GGA2 | As recent studies have shown GGA1 and GGA3 protein level alterations in postmortem samples of AD patients, we also compared the expression of GGA2 in 26 temporal lobe samples obtained from control and AD patients. Some AD patients showed altered GGA2 levels compared with matched controls. (D) GGA2 levels were analyzed by Western blot in postmortem temporal lobe samples of AD and controlled patients (MADRC). | Protein level |
| 97 | NBS1 | To test whether NBS1 indeed interacts with the mTOR/Rictor/SIN1 complex, co-immunoprecipitation assays using extracts from a lung cancer cell line H1299 were used. | Protein level |
| 43 | prostatic acid phosphatase | In semen, proteolytic peptide fragments from prostatic acid phosphatase can form amyloid fibrils termed SEVI (semen-derived enhancer of viral infection). | Protein level |
| 43 | FIT2 | Miranda etAI-nonASCII- al4 found that mice with adiposeaI-nonASCII-A-nonASCII-A<90>specific FIT2 deficiency developed severe, progressive lipodystrophy with fatty liver, tissue macrophage infiltration, and insulin resistance, with few but abnormally large lipid droplets on histology. | Metabolite level |
| 19 | CCDC3 | Thus, we decided first to determine if CCDC3 could affect lipid metabolism in hepatic cancer cells by performing a metabolomics analysis. | Metabolite level |
| 108 | alpha-N-acetylgalactosaminidase | The degradation of blood group glycolipid A-6-2 (GalNAc(alpha1–>3) [Fuc alpha1–>2]Gal(beta1–>4)GlcNAc(beta1–>3)Gal(beta1–>4)Glc(beta1–>1')C er, IV2-alpha-fucosyl-IV3-alpha-N-acetylgalactosaminylneolact otetraosylceramide), tritium-labeled in its ceramide moiety, was studied in situ, in skin fibroblast cultures from normal controls, from patients with defects of lysosomal alpha-N-acetylgalactosaminidase, and patients with other lysosomal storage diseases. | Metabolite level |
| 84 | NNMT | To address this problem, we employed an untargeted metabolomics approach18, where metabolomes from NNMT-OE and GFP-OE renal carcinoma (769P), ovarian cancer (OVCAR3), and melanoma (MUM2C) cells were comparatively analyzed by an HPLC-Q-TOF-MS system operating in the broad mass scanning mode (m/z range of 50aI-nonASCII-A-nonASCII-A-nonASCII-1200 Da). | Metabolite level |
| 167 | DDHD2 | In line with the function of DDHD2 in lipid metabolism and its role in the CNS, an abnormal lipid peak indicating accumulation of lipids was detected with cerebral magnetic resonance spectroscopy, which provides an applicable diagnostic biomarker that can distinguish the DDHD2 phenotype from other complex HSP phenotypes. | Metabolite level |

Also, during the manual review, we determined additional characteristics of the text fragments and terms by answering the following questions:

- Does the text fragment allow us to judge the category of the experimental method?
- Are highlighted terms sufficient to judge the category of experimental method?

The results for the twenty terms from DNA/RNA, proteins and metabolites categories associated with the biggest prevalence of positive (upper ten terms on the Figure) and negative (lower ten) answers are given in Figures 5–7.

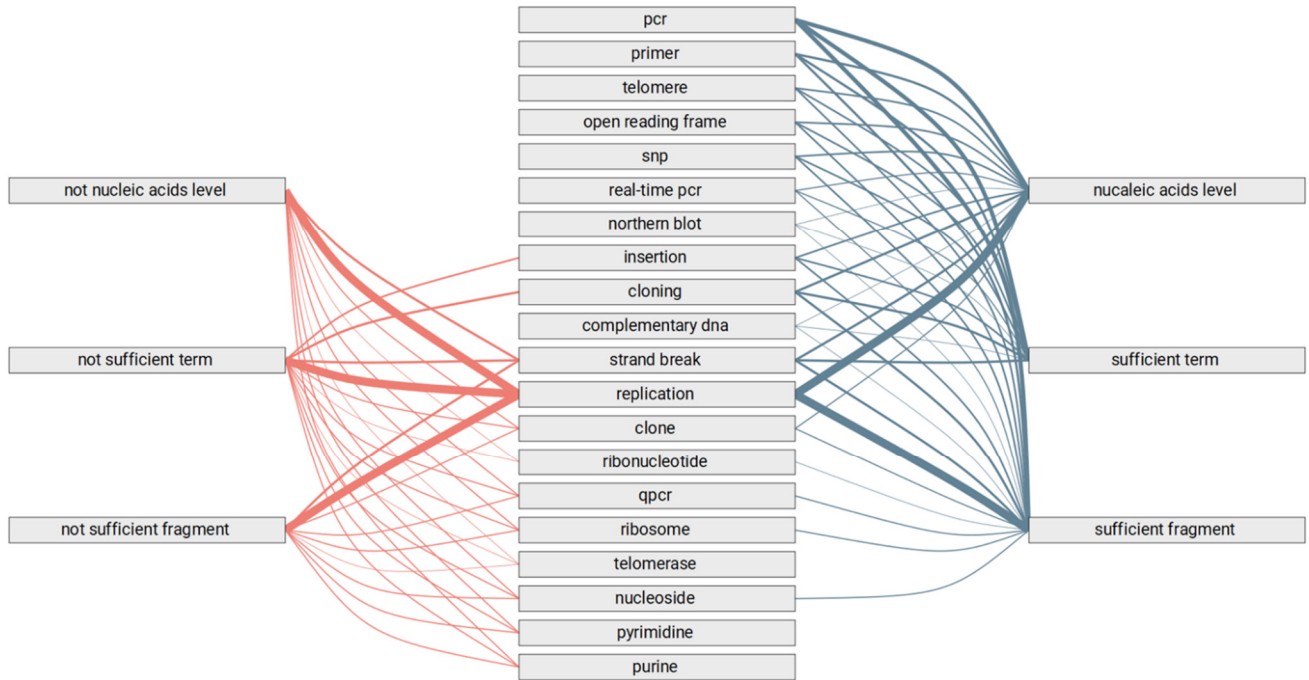

**Figure 5.** Association between the terms from the DNA/RNA category of the dictionary and answers obtained during the manual review. Thicker lines connecting the term and an answer mean more cases. Blue lines connect terms and positive answers. Red lines connect the terms and negative answers.

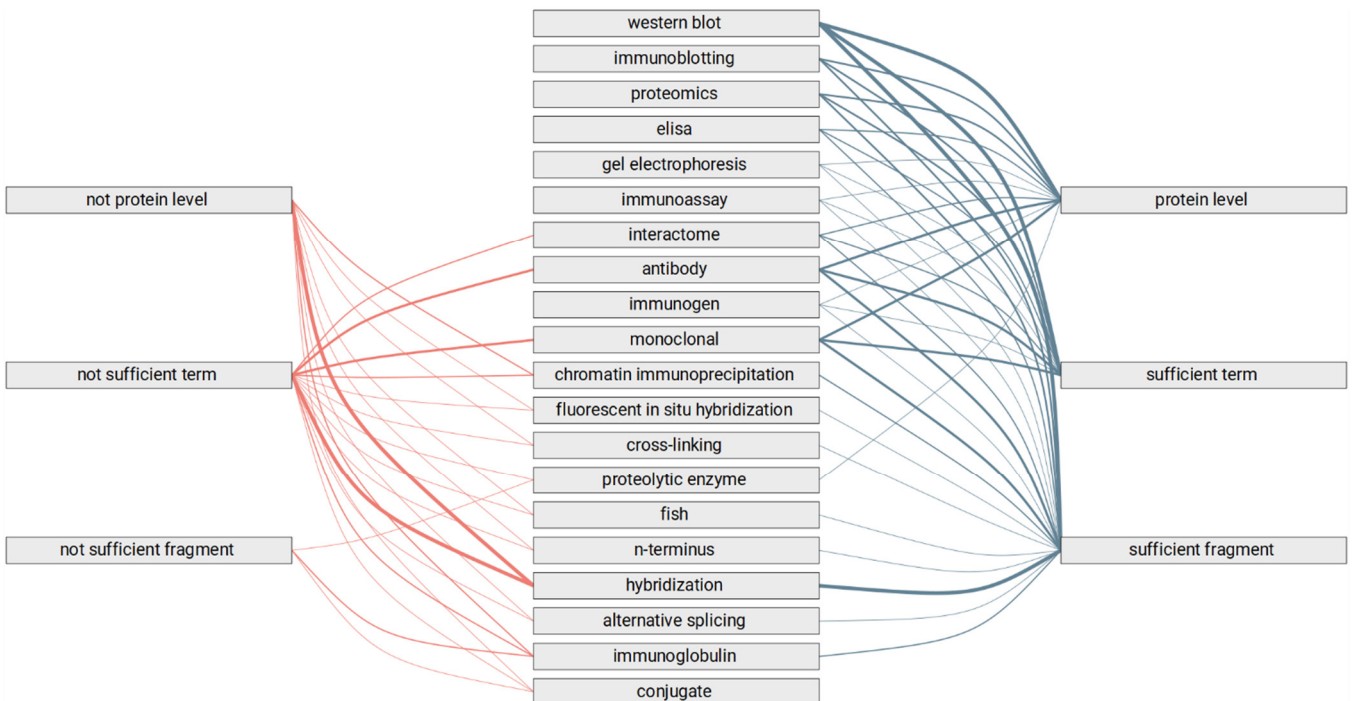

**Figure 6.** Association between the terms from the protein category of the dictionary and answers obtained during the manual review. Thicker lines connecting the term and an answer mean more cases. Blue lines connect terms and positive answers. Red lines connect the terms and negative answers.

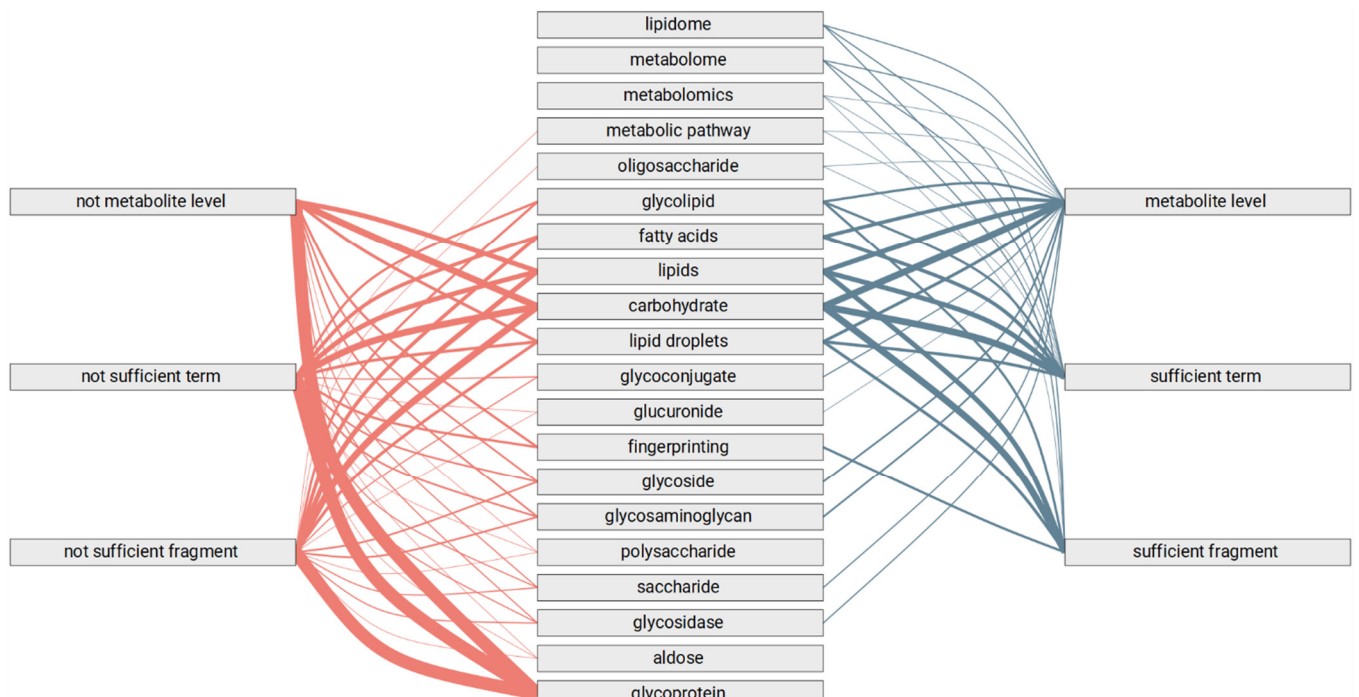

**Figure 7.** Association between the terms from the metabolite category of the dictionary and answers obtained during the manual review. Thicker lines connecting the term and an answer mean more cases. Blue lines connect terms and positive answers. Red lines connect the terms and negative answers.

Figure 5 shows that the intersection between the terms associated with the prevailing positive answers and terms for which association with the negative answer prevails is small. Probably only the term "replication" is associated with many positive and negative answers. It could be considered in further studies, which will benefit from the optimized dictionary. This Figure shows that the dictionary could be optimized by excluding the bottom-rated terms without a significant decrease in the sensitivity of the screening.

From Figure 6, it can be seen that in comparison with the DNA/RNA category protein category has a more intertwined dictionary: many of the terms primarily associated with the negative answers are also associated with the positive ones. It makes further optimising the terms for the protein level of study harder since it will not be possible to delete bottom-rated terms without sacrificing the sensitivity of the dictionary-based screening.

As one can see from Figure 7, there are only three terms specific to the metabolite level of study (lipidome, metabolome, metabolomics) in the dictionary. All other terms are associated with negative answers too. The other important observation is that terms mainly describe some metabolites. Thus, to be categorized as related to the metabolites level of studies, the text fragment should include the name of the additional biological entity. This indicates the possible limitation of our systematic approach: such an additional entity is not required to judge the belonging of text fragments to DNA/RNA or protein level. Results related to the metabolite level are also interesting because, unlike the results for DNA/RNA and protein level, they show that there are such terms in our dictionary, which could be associated with both negative and positive answers ("lipids", "carbohydrate", "lipid droplets" are the most prominent examples). This additionally indicates the peculiarities of the dictionary-based screening for the text fragments associated with the metabolite level of study.

Overall, the manual review of the text fragments allowed us to precisely map the small fraction (600) of text fragments to the DNA/RNA, protein, or metabolite level of study. These results could be of value on their own or be used as the training sets for the automated review conducted using machine learning techniques, the latter path we choose. Moreover, the analysis of these results, combined with the associated terms from

the dictionary used during the screening stage, allowed us to highlight the ways to optimise the screening stage for further studies.

### 3.3. Automated Review of the Text Fragments

The results of the machine learning review of the 31,260 retrieved text fragments are summarized in Figure 8.

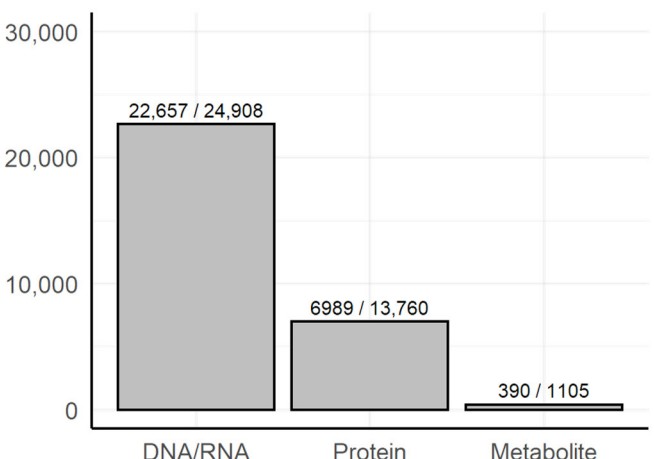

**Figure 8.** Results of the automated review of all included text fragments in each dictionary-defined category. The total numbers of the text fragment in the dictionary-defined categories are given after the slash.

According to Figure 8 obtained results are consistent with the manual classification: from about 9 to 65 percent of the text fragments were classified as unrelated to the categories defined using a dictionary. Moreover, the downward trend in the number of text fragments in the series DNA/RNA → proteins → metabolites remain here (see Figures 3, 4 and 8). The results of the automated mapping may be found in the Supplementary Materials (see Table S3).

Overall, the results of the mapping indicate the prevalence of the association studies between biological macromolecules and diseases conducted on the level of nucleic acids (DNA or RNA) over the protein and metabolite levels of the study. Probably, there are the following main factors contributing to this situation:

- Methods of genomics and transcriptomics are still the most accessible and mature ones. They allow monitoring of the molecules both in high-throughput and targeted modes. Thus, such methods are the most often used.
- Despite the start of the postgenomic era and recent progress in proteomics [28] and metabolomics [29], the number of scientific papers on genes and their transcripts accumulated in the public domain remains unmatched.
- The number of biological macromolecules associated with diseases via metabolites is limited (the macromolecule should be principally related to some metabolite and studied enough for this relationship to be known).

Our results suggest that given the number of genomic and transcriptomic evidence of the role of distinct biological macromolecules in pathology, there is a clear need to translate them into evidence on the level of proteins and metabolites [2,8]. Furthermore, since genes and their transcripts rarely act in the cell by themselves, more direct evidence may be of greater value for the basic and applied studies.

### 4. Conclusions

Our systematic mapping review of the text fragments from Open Targets describing relations between biological macromolecules and diseases has shown the prevalence of the knowledge obtained while studying the DNA/RNA variants and levels. Therefore,

translating such knowledge on the level of proteins and metabolites may be proposed to improve the reliability of the association between potential therapeutic targets and diseases.

**Supplementary Materials:** The following supporting information can be downloaded at: https://www.mdpi.com/article/10.3390/cimb45040223/s1, Table S1: Dictionary; Table S2: Manual Mapping Review; Table S3: Automated Mapping Review; File S1: questionary (DNA/RNA category); File S2: questionary (protein category); File S3: questionary (metabolites category).

**Author Contributions:** Conceptualization, P.V.P., O.I.K. and E.V.I.; methodology, P.V.P.; manual review of the texts' fragments, P.V.P., O.I.K. and E.V.I.; software, P.V.P.; writing—original draft preparation, P.V.P.; writing—review and editing, P.V.P., O.I.K. and E.V.I.; visualization, P.V.P. All authors have read and agreed to the published version of the manuscript.

**Funding:** The work was performed within the framework of the Program for Basic Research in the Russian Federation for a long-term period (2021–2030) (122030100170-5).

**Institutional Review Board Statement:** Not applicable.

**Informed Consent Statement:** Not applicable.

**Data Availability Statement:** The data supporting the results of this study can be found in the Supplementary Materials.

**Conflicts of Interest:** The authors declare no conflict of interest.

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
