# Peer review of "Identification of Potential Therapeutic Targets on the Level of DNA/mRNAs, Proteins and Metabolites: A Systematic Mapping Review of Scientific Texts’ Fragments from Open Targets"

_cimb, doi:10.3390/cimb45040223_

Round 1

Reviewer 1 Report

Comments and Suggestions for Authors

In this manuscript, the author mapped the biological macromolecules from Open Targets and distinguished whether they belong to nucleic acids, proteins or metabolites. The article is well-written, but I didn't see the usage of this article. The authors claimed that they found the association between diseases and macromolecules, but the data cannot be found. Rejection is suggested.

Comments:

1. In Figure 1, there are two dotted lines towards "fragments included in automated review." As illustrated in the Materials, classifiers were built using the results of the manual classification, so it illustrates the dotted line between "manual review" and "automated review." Could the authors provide more illustrations of the other dotted line? Why were the excluded records also included in the automated review?

2. There is a typo in line 200. There is a space between "31" and "260."

3. In lines 209-213, there are no references. Please consider adding references to support the statement.

4. In the keywords, the authors included "target-disease association." But as presented in Figures 5, 6, 7, and 8, the targets were not associated with disease. Please consider adding data or removing "target-disease association" from the keyword.

Reviewer 2 Report

Comments and Suggestions for Authors

An interesting manuscript. 

Recommendation for the authors.

1. In order to gain more scientific audience, the manuscript should be composed with more informative or with more examples biomarkers-diseases in addition to the given table. In the present form there is more general texts of the main findings.

2. Please give some discussion exploring the following: advantages and disadvantages of the Open Target platform, usefulness in the drug development.

3. Make comparison to other similar platforms and other levels of biomarkers, if applicable. 

4. References list should be corrected in terms of guidelines for the authors.

Round 2

Reviewer 1 Report

Comments and Suggestions for Authors

The author addressed each comment. Acceptance is suggested.